# X-MoGe: A Cross-Modal Adaptation Framework with Mixture-of-Experts and Geometry Guidance for Heterogeneous Collaborative Perception

Wenkai Lin [1 2]   Zhihong Liu [1 2]   Chenglu Wen [1 2]

## Abstract

Multi-agent collaborative perception improves perception range and robustness in autonomous driving. However, most existing methods assume homogeneous sensors and perception networks, which is unrealistic in real-world heterogeneous systems. Differences in sensing modalities and independently trained models lead to significant semantic and geometric inconsistencies, limiting effective collaboration. To solve these problems, we propose a novel cross-modal adaptation framework with Mixture-of-Experts and geometry-guided fusion for heterogeneous collaborative perception, named X-MoGe. Specifically, we propose a Pixel-level Mixture-of-Experts (P-MoE) module, which adaptively models modality-specific semantic characteristics under heterogeneous sensing conditions. In addition, a geometry-guided feature fusion module incorporates explicit geometric priors to enforce spatial alignment and consistency in the BEV space. Extensive experiments on OPV2V and DAIR-V2X datasets demonstrate that the proposed method achieves state-of-the-art performance in heterogeneous collaborative perception.

## 1. Introduction

With the rapid development of autonomous driving, multi-agent collaborative perception has emerged as an effective paradigm to overcome the limitations of single-agent perception by sharing information among agents, thereby extending perception range and improving robustness under limited viewpoints or occlusions.

[1]Fujian Key Laboratory of Urban Intelligent Sensing and Computing, Xiamen University, 361005, P.R. China [2]Key Laboratory of Multimedia Trusted Perception and Efficient Computing, Ministry of Education of China, Xiamen University, 361005, P.R. China.. Correspondence to: Chenglu Wen <clwen@xmu.edu.cn>.

*Proceedings of the 43rd International Conference on Machine Learning*, Seoul, South Korea. PMLR 306, 2026. Copyright 2026 by the author(s).

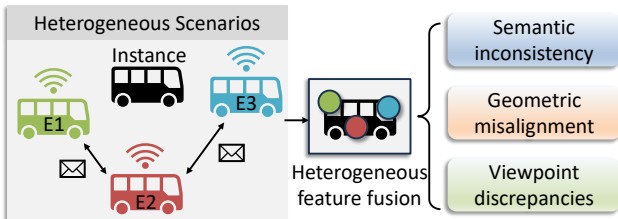

*Figure 1.* In real-world autonomous driving scenarios, different agents are often equipped with diverse sensors and different perception encoders (E1, 2, 3), resulting in significant heterogeneity in the generated feature representations across agents. Such heterogeneity introduces substantial challenges for collaborative perception.

In multi-agent collaborative perception, existing methods (Chen et al., 2019; Xu et al., 2022e;b; Hu et al., 2024; Zhang et al., 2024; Song et al., 2025) are mostly developed under homogeneous assumptions, where agents share similar sensor configurations and perception models. In contrast, real-world systems are inherently heterogeneous, as vehicles from different platforms and manufacturers adopt diverse sensor suites (e.g., LiDAR, cameras, or multi-modal sensors) and independently trained perception networks. As shown in Figure 1 and (a), (b), and (c) of Figure 2, large gaps remain across modalities in the BEV space, including semantic inconsistency, geometric misalignment, and viewpoint discrepancies, which significantly hinder effective heterogeneous feature fusion.

To address heterogeneous collaborative perception, several studies (Xu et al., 2022a; Xiang et al., 2023; Luo et al., 2024; Wang et al., 2025) have explored cross-modal or cross-network feature interaction mechanisms. However, research on multi-agent cross-modal feature interpreters is still an open and underexplored area. Existing methods (Lu et al., 2024; Xia et al., 2025) typically rely on unified training networks or a single feature interpretation strategy, without explicitly accounting for inherent discrepancies of different modalities. These discrepancies often manifest as significant geometric and semantic differences across agents. As shown in Figure 2, different sensing modalities exhibit uneven feature stability across channels and varying sensitivity to spatial perturbations and data variations. Moreover, features extracted from heterogeneous sensors

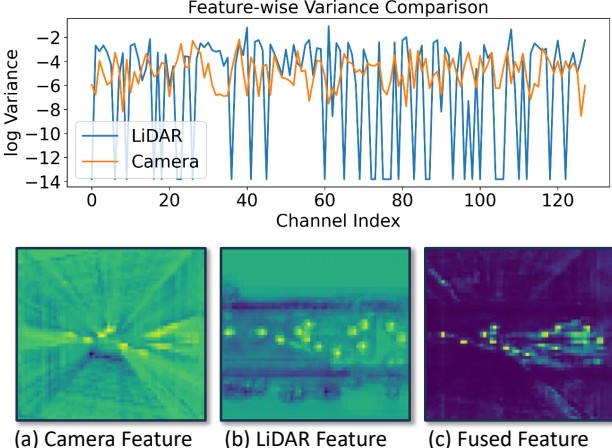

(a) Camera Feature    (b) LiDAR Feature    (c) Fused Feature

*Figure 2.* By computing channel-wise variance, we assess feature stability under data and spatial variations: channels with low variance tend to encode stable geometric patterns, while channels with high variance are more affected by sensing noise and appearance changes. (O'shea & Nash, 2015; Yamashita et al., 2018) (a) and (b) show camera and LiDAR BEV features from PointPillars (Lang et al., 2019) and ResNet50 (Koonce, 2021b) + Lift-Splat (Philion & Fidler, 2020), and (c) shows the ViT-fused feature, where substantial noise is introduced.

encode geometric information and semantic cues under different fields of view, leading to modality-dependent noise amplification when directly projected or fused into the BEV space. Such discrepancies not only hinder spatial alignment but also introduce unstable feature responses, ultimately degrading collaborative perception performance.

Therefore, to better achieve semantic and geometric feature alignment across different modalities and viewpoints, we propose X-MoGe, a cross-modal adaptation framework with Pixel-level Mixture-of-Experts (P-MoE) and geometry-guided feature fusion for heterogeneous multi-agent collaborative object detection, as illustrated in Figure 3. The proposed P-MoE module performs pixel-wise, modality-aware expert selection to interpret heterogeneous features, enabling effective handling of semantic diversity and uncertainty under heterogeneous sensing conditions, while the proposed geometry-guided feature fusion module considers the geometric consistency of different modalities and is composed of an Uncertainty-Aware head, Sobel-based Gradient Encoding (SGE), and Edge-Aware Enhancement (EAE).

Specifically, the features of different agents are first adaptively processed using P-MoE. Then, the uncertainty-aware head estimates adapted feature reliability to modulate cross-agent contributions, SGE explicitly captures geometric structures and spatial gradients in the BEV space, and EAE further enhances boundary-aware geometric cues during fusion. By jointly integrating semantic adaptation and geometry-

guided alignment, the proposed framework effectively exploits complementary semantic and geometric information across heterogeneous modalities.

To evaluate our method, extensive experiments on OPV2V (Xu et al., 2022e) and DAIR-V2X (Yu et al., 2022) datasets demonstrate that our method achieves state-of-the-art performance in heterogeneous collaborative object detection while exhibiting strong robustness and scalability. Our main contributions are summarized as follows:

- We propose a novel multi-agent cross-modal adaptation framework for heterogeneous collaborative perception named X-MoGe that integrates multi-agent image and LiDAR features into a shared cross-modal representation.

- We integrate a Pixel-level Mixture-of-Experts (P-MoE) module for adaptive semantic reasoning with a geometry-guided feature fusion module that enforces BEV-space alignment, improving cross-modal consistency under heterogeneous multi-agent sensing.

- Our method achieves superior performance across both the OPV2V and DAIR-V2X datasets.

## 2. Related Work

### 2.1. Multi-Agent Collaborative Perception

Multi-agent collaborative perception has attracted increasing attention in autonomous driving due to its potential to extend perception range and improve robustness through information sharing among connected agents. Early works (Chen et al., 2019; Xu et al., 2022b;d; Hu et al., 2024; Zhang et al., 2024; Song et al., 2025) typically compress features into lightweight BEV representations for transmission to reduce the communication bandwidth among agents and primarily focus on homogeneous collaborative perception, where all agents are assumed to share identical sensing modalities and perception architectures. However, in real-world autonomous driving scenarios, different agents often suffer from sensor heterogeneity or perception model heterogeneity, which lead to substantial discrepancies in feature distributions and significantly challenge effective collaborative perception.

### 2.2. Heterogeneous Feature Fusion

In recent years, more and more researchers have focused on mitigating heterogeneity issues in multi-agent collaborative perception to improve effectiveness. MPDA (Xu et al., 2022a), PnPDA (Luo et al., 2024), and Polyinter (Xia et al., 2025) propose heterogeneous feature interpreters to address the domain differences between different encoders across multiple agents. HM-ViT (Xiang et al., 2023), HEAL (Lu

et al., 2024), and CoCMT (Wang et al., 2025) introduce cross-modal fusion networks that learn a unified feature representation across different modalities through a single training network. However, existing interpreters and fusion methods do not sufficiently account for the semantic and geometric misalignment that arises in cross-modal feature fusion due to differences in sensing viewpoints and spatial representations across modalities. As a result, feature correspondences remain incomplete, which limits the ability of current methods to fully exploit the complementary information among heterogeneous agents.

### 2.3. Mixture of Experts

Mixture-of-Experts (MoE) was originally proposed to enable conditional computation through multiple expert sub-networks and a gating mechanism, thereby increasing model capacity while controlling computational cost. In the NLP domain, prior works (Lepikhin et al., 2020; Fedus et al., 2022) have leveraged MoE to capture textual variations; similarly, MoE has demonstrated strong performance in image classification (Riquelme et al., 2021), video understanding (Fan et al., 2022), multimodal learning, and multi-view perception. In multi-agent collaborative perception, different agents often adopt heterogeneous sensor configurations and encoder architectures, resulting in significant discrepancies in both semantic and geometric feature spaces. CoBEVMoE (Kong et al., 2025) introduces a dynamic MoE framework to explicitly model both the similarity and diversity of multi-agent observations. However, it only achieves coarse-grained alignment and does not explicitly account for spatial discrepancies. In contrast, we introduce MoE from the perspective of pixel-level feature interpretation, enabling finer-grained alignment of features from different modalities and agents within a shared representation space, and providing a consistent semantic foundation for subsequent geometry-guided fusion.

## 3. Method

### 3.1. Problem Formulation

We consider a heterogeneous collaborative perception system consisting of multiple agents equipped with different sensing modalities, such as LiDAR and camera. Each agent independently captures observations from its local viewpoint and extracts intermediate feature representations using modality-specific encoders.

Formally, let $\mathcal{A} = \{1, \ldots, N\}$ denote the set of agents. For agent $i$, its raw sensory input $x_i^m$ from modality $m$ is transformed into a BEV feature map $\mathbf{F}_i^m \in \mathbb{R}^{C \times H \times W}$, where $C$, $H$, and $W$ denote the channel dimension and spatial resolution, respectively. Due to heterogeneous sensing conditions and modality gaps, these features exhibit significant discrep-

ancies in semantic distributions and geometric properties.

Our goal is to learn a unified cross-modal adaptation and fusion framework that (1) mitigates modality-specific semantic bias at the feature level, and (2) enables robust cooperative perception through geometry-consistent feature fusion across agents. Our framework is illustrated in Figure 3.

### 3.2. Heterogeneous Feature Adaptor

**Pixel-level Mixture of Experts (P-MoE).**

To adapt heterogeneous features at a fine-grained level, we propose a Pixel-level Mixture of Experts (P-MoE) module as illustrated in Figure 4, which performs expert routing and feature transformation independently at each spatial location. This design enables spatially adaptive cross-modal semantic alignment while preserving dense geometric correspondence.

Given an input feature map $\mathbf{F} \in \mathbb{R}^{B \times C \times H \times W}$, P-MoE first reshapes the feature into a set of pixel-wise embeddings:

$$\mathbf{f}_{b,h,w} \in \mathbb{R}^C, \quad b \in [1, B], \ h \in [1, H], \ w \in [1, W] \quad (1)$$

Each pixel embedding is then independently routed to a set of $K$ expert networks through a gating function.

Specifically, we employ two gating networks with identical structures but independent parameters, corresponding to ego-agent features and non-ego agent features, respectively. For a given pixel embedding $\mathbf{f}$, the gating weights are computed as:

$$\mathcal{M} = \begin{cases} \mathrm{Softmax}(G_{\mathrm{ego}}(\mathbf{f})), & \text{ego agent} \\ \mathrm{Softmax}(G_{\mathrm{non}}(\mathbf{f})), & \text{non-ego agent} \end{cases} \quad (2)$$

where $\mathcal{M} \in \mathbb{R}^K$ denotes the expert assignment scores. The expert set consists of $K$ lightweight MLPs, each modeling a distinct semantic transformation:

$$E_k(\mathbf{f}) = W_k^{(2)} \cdot \mathrm{ReLU}\left(W_k^{(1)}\mathbf{f}\right), \quad k = 1, \ldots, K \quad (3)$$

All experts share the same architecture but maintain independent parameters. The output feature at each pixel is computed as a soft aggregation of expert outputs:

$$\tilde{\mathbf{f}} = \sum_{k=1}^{K} \mathcal{M}_k \cdot E_k(\mathbf{f}) \quad (4)$$

Finally, the pixel-wise adapted features are reshaped back to the original spatial layout, yielding the adapted feature map $\tilde{\mathbf{F}} \in \mathbb{R}^{B \times C' \times H \times W}$. By performing expert selection at the pixel level and employing modality-aware gating functions, P-MOE adaptively decides, at each spatial location, how to

**(a) P-MoE Adaptor**

Ego Agent → Encoder → BEV Feature → P-MoE → Adapted Feature → Fusion → Decoder → Detection Output

Neb Agent → Encoder → BEV Feature

**(b) Geometry-Guided Feature Fusion**

LiDAR → Encoder → Feature $F_1$ → $\alpha$ → Geometry-Guided Refinement → SGE → EAE → Refined Feature

Multi-view Images → Encoder → Feature $F_2$ → $1-\alpha$ → $\oplus$ → $F_{fused}$ → Conv2d → ReLU → Conv2d → LN → Spatial Cross-Attention Transformer

UA-head → $\alpha$

Fused BEV Feature $LN\big(\phi\big(\text{Re}LU(\phi(F_{fused}))\big)\big)$

*Figure 3.* The overall architecture of the proposed framework. This framework consists of two main components: (a) Mixture-of-Experts (MoE) adaptor, which performs pixel-level cross-modal adaptation to account for heterogeneous semantic characteristics under different sensing modalities, and (b) Geometry-guided feature fusion module, which integrates multi-agent features by explicitly incorporating geometric cues to support spatially consistent collaborative perception. *P-MOE*: Pixel-level Mixture-of-Experts; *SGE*: Sobel-based Gradient Encoding; *EAE*: Edge-Aware Enhancement; *UA-head*: Uncertenty-Aware head; $\alpha$: pixel-level confidence; $\phi$: conv2d;

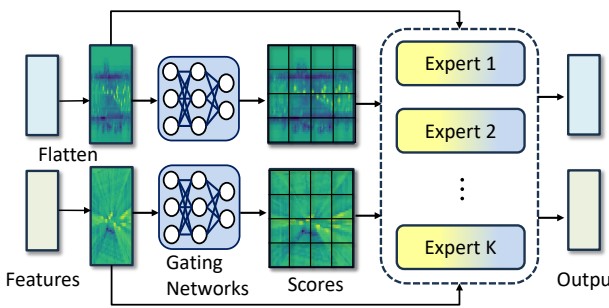

*Figure 4.* The architecture of P-MoE, which performs fine-grained feature adaptation at each modality and spatial location.

fuse image and LiDAR features by emphasizing LiDAR in structurally salient regions, images in texture-rich areas, and expert-driven fusion where both modalities are weak.

### 3.3. Geometry-Guided Feature Fusion

**Uncertainty-Aware Fusion.**

In heterogeneous collaborative perception, features from different agents may exhibit varying reliability due to sensing noise, viewpoint changes, or modality-specific limitations. To explicitly account for such uncertainty during cooperative fusion, we introduce an uncertainty-aware fusion mechanism that adaptively balances contributions from ego and neighboring agents at the pixel level.

Given the feature map $\mathbf{F}_{\text{cav}} \in \mathbb{R}^{B \times C \times H \times W}$ from a collaborating agent, we employ a lightweight convolutional head to predict a pixel-wise uncertainty map. Specifically, the uncertainty head consists of two convolutional layers with a ReLU activation in between, and outputs a single-channel log-variance map:

$$\mathbf{U} = h_{\text{uncert}}(\mathbf{F}_{\text{cav}}), \quad \mathbf{U} \in \mathbb{R}^{B \times 1 \times H \times W} \tag{5}$$

The predicted log-variance is converted into a precision-like confidence measure through an exponential transformation:

$$\mathbf{P} = \exp\left(-\text{clip}(\mathbf{U})\right) \tag{6}$$

where the clipping operation stabilizes training by preventing extreme values. We further normalize the precision to obtain a bounded fusion weight:

$$\boldsymbol{\alpha} = \frac{1}{\mathbf{P} + 1} \tag{7}$$

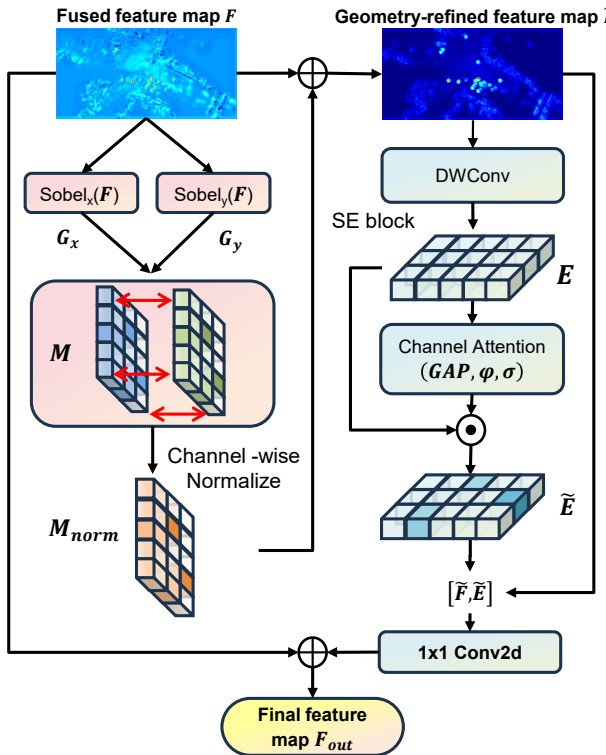

**Figure 5.** The workflow diagram of the geometry-guided refinement module. The left area represents the Sobel-based Gradient Encoding (SGE) block, while the right denotes the Edge-Aware Enhancement (EAE) block. The final fused features preserve the intrinsic characteristics of the original point cloud while enhancing the geometric expressiveness of the original features.

where $\alpha \in (0, 1)$ determines the degree of pixel-wise contribution of different agents. The final fused feature is computed as a convex combination of the collaborating agent feature and the ego feature:

$$\mathbf{F}_{\text{fused}} = (1 - \boldsymbol{\alpha}) \odot \mathbf{F}_{\text{cav}} + \boldsymbol{\alpha} \odot \mathbf{F}_{\text{ego}} \qquad (8)$$

Then, the fused feature $\mathbf{F}_{\text{fused}}$ is further refined by sequentially applying a convolution, ReLU activation, another convolution, and Layer Normalization, as formulated below:

$$\mathbf{F}'_{fused} = \mathcal{LN}\left(\phi\left(\text{ReLU}\left(\phi\left(\mathbf{F}_{fused}\right)\right)\right)\right) \qquad (9)$$

The refined $F'_{fused}$ is subsequently integrated with the ego agent's features processed by Geometry-Guided Refinement through a Spatial Cross-Attention operation.

**Geometry-Guided Refinement.**

While uncertainty-aware fusion dynamically balances features from different agents, it does not explicitly emphasize geometric structures that are critical for spatial reasoning. To enhance geometry consistency and structural awareness, we introduce a geometry-guided refinement module that injects local geometric cues into the fused feature representation, which consists of a Sobel-based Gradient Encoding (SGE) for structural refinement, followed by an Edge-Aware Enhancement (EAE) block. It is worth noting that, due to the strong geometric consistency of the LiDAR modality, we apply the Geometry-Guided Refinement operation only to the LiDAR features, which are then used to guide better alignment of the image modality.

Specifically, given a feature map $\mathbf{F} \in \mathbb{R}^{B \times C \times H \times W}$, we estimate geometry cues by computing the spatial gradient magnitude using Sobel (Gao et al., 2010) operators. Specifically, depthwise Sobel convolutions are applied independently to each channel to obtain horizontal and vertical gradients:

$$\mathbf{G}_x = \text{Sobel}_x(\mathbf{F}), \quad \mathbf{G}_y = \text{Sobel}_y(\mathbf{F}) \qquad (10)$$

The per-channel gradient magnitudes are then aggregated across channels to form a single geometry response map:

$$\mathbf{M}(h, w) = \sqrt{\sum_{c=1}^{C} \left(G_x^c(h, w)^2 + G_y^c(h, w)^2\right)} \qquad (11)$$

To ensure numerical stability and consistency across samples, the geometry response is normalized to $[0, 1]$ on a per-sample basis. The resulting normalized geometry map $\mathbf{M}_{\text{norm}} \in \mathbb{R}^{B \times 1 \times H \times W}$ reflects the strength of local structural variations, such as object boundaries and depth discontinuities.

Instead of concatenating geometry features and increasing channel dimensionality, we adopt an additive refinement strategy. The normalized geometry map is broadcast along the channel dimension and directly added to the fused feature:

$$\tilde{\mathbf{F}} = \mathbf{F} + \mathbf{M}_{\text{norm}} \qquad (12)$$

This additive formulation preserves the original feature dimensionality and acts as a geometry-aware bias term that selectively enhances spatial locations with strong structural responses. By emphasizing geometrically salient regions while maintaining feature compatibility with subsequent modules, the proposed geometry-guided refinement improves spatial consistency and robustness without introducing additional parameters or computational overhead.

In addition, to enhance geometric boundary awareness in cooperative features, we introduce an Edge-Aware Enhancement (EAE) module that explicitly models local edge responses and selectively integrates them with semantic representations. We first apply a depthwise convolution to extract channel-wise edge features:

$$\mathbf{E} = \text{DWConv}(\tilde{\mathbf{F}}) \qquad (13)$$

where the depthwise operation preserves modality-specific semantics while capturing local spatial gradients related to object boundaries.

To adaptively emphasize informative edge channels and suppress noisy responses, a channel attention mechanism is applied to the edge feature:

$$\tilde{\mathbf{E}} = \mathbf{E} \odot \sigma\big(\varphi(\text{GAP}(\mathbf{E}))\big) \qquad (14)$$

where $\text{GAP}(\cdot)$ denotes global average pooling, $\varphi(\cdot)$ represents a lightweight squeeze-and-excitation transformation, $\sigma(\cdot)$ is the sigmoid activation, and $\odot$ denotes channel-wise multiplication. The original feature and the enhanced edge feature are then concatenated and fused through a $1 \times 1$ convolution:

$$\mathbf{F}' = \text{Conv}_{1 \times 1}([\tilde{\mathbf{F}}, \tilde{\mathbf{E}}]) \qquad (15)$$

followed by a residual connection to preserve the original semantic information:

$$\mathbf{F}_{\text{out}} = \tilde{\mathbf{F}} + \mathbf{F}' \qquad (16)$$

Geometry-Guided Refinement enables effective integration of geometric cues, improving boundary sensitivity while maintaining stable optimization and feature consistency across heterogeneous agents.

**Spatial Cross-Attention Module.**

The spatial cross-attention module is implemented based on the fused axial attention mechanism (Xu et al., 2022c) for modeling cross-agent spatial interactions. Spatial cross-attention is applied in two successive stages, where attention is first computed within partitioned local regions to capture fine-grained spatial correspondences, followed by a second stage operating on global spatial regions to incorporate broader spatial dependencies across agents. This formulation enables efficient aggregation of long-range spatial context while maintaining computational efficiency.

**Decoder and Loss.**

The refined cooperative feature is decoded into classification scores and 3D bounding box predictions, where each box is parameterized as $(x, y, z, h, w, l, \theta)$. Smooth L1 loss is used for bounding box regression to ensure stable optimization of geometric parameters, while Focal Loss (Lin et al., 2020) is adopted for classification to mitigate foreground–background class imbalance. The network is trained end-to-end using a weighted combination of the regression and classification losses.

## 4. Experiments

### 4.1. Dataset and Evaluation Metrics

We conducted experiments on the OPV2V (Xu et al., 2022e) and DAIR-V2X (Yu et al., 2022) datasets. OPV2V is a large-scale public simulation dataset designed for vehicle-to-vehicle (V2V) autonomous driving scenarios. It contains 73 scenes across 6 road types and 9 cities, comprising approximately 12K frames of LiDAR point clouds and RGB

| Encoder | Voxel Resolution | 2D/3D CNN layers | Half L/C Range (x,y) |
|---------|------------------|------------------|----------------------|
| pp8 | 0.8, 0.8, 4 | 10 / 0 | 140.8, 40 |
| pp4 | 0.4, 0.4, 4 | 19 / 0 | 140.8, 40 |
| vn4 | 0.4, 0.4, 0.4 | 0 / 3 | 140.8, 40 |
| res50 | 0.4, 0.4, 20 | 50 / 4 | 102.4, 51.2 |
| eff | 0.4, 0.4, 20 | 16 / 4 | 102.4, 51.2 |

*Table 1.* Characteristics of Different Encoders. pp8 and pp4 denote PointPillars (Lang et al., 2019) encoders, vn4 denotes a VoxelNet (Zhou & Tuzel, 2018) encoder, while res50 represents ResNet-50 (Koonce, 2021b) and eff denotes EfficientNet (Koonce, 2021a).

camera images, along with over 230K annotated 3D bounding boxes, covering a wide range of heterogeneous scenarios. DAIR-V2X is the first large-scale real-world dataset for vehicle–infrastructure cooperative autonomous driving, featuring multi-modal and multi-view data collected from vehicle-side cameras, vehicle-side LiDARs, infrastructure-side cameras, and infrastructure-side LiDARs. It encompasses diverse environmental conditions, including sunny, rainy, and foggy weather, daytime and nighttime settings, as well as urban roads and highways. We evaluate model performance using mean Average Precision (mAP) under IoU thresholds of 0.3, 0.5, and 0.7.

### 4.2. Implementation Details

In Table 1, we follow the experimental settings of previous works (Lu et al., 2024; Xia et al., 2025). Pp8, pp4, and vn4 denote encoders for the LiDAR modality, while Lift-Splat (Philion & Fidler, 2020) with res50 and eff represent encoders for the image modality. LiDAR encoders (pp8, pp4, vn4) rely on 3D CNN layers, use voxel resolutions of 0.8 or 0.4 and cover a large perception range of about 140.8×40 m. In contrast, image encoders (res50, eff) rely on 2D CNN layers, produce 20 BEV height representations, and operate over a smaller range of 102.4×51.2 m. And the two modalities employ different numbers of 2D/3D CNN layers. These numerical differences highlight inherent gaps in heterogeneous feature fusion.

All experiments are conducted on the OPV2V and DAIR-V2X datasets following the heterogeneous collaborative perception setting (Xia et al., 2025). During training, we use both LiDAR and camera inputs, where LiDAR-based predictions are treated as supervision signals.

In Table 2, all collaborative agents' encoders and the ego agent's detection head are frozen and inaccessible while testing under different heterogeneous scenarios. while in Table 3, we conduct end-to-end training (Lu et al., 2024) on both the OPV2V and DAIR-V2X datasets. For OPV2V, we fix the ego agent to use pp8, and configure the ego–neb1–neb2 agents as pp8-res50-vn4 and pp8-eff-vn4. During training, the ego agent randomly selects the encoder of the neighboring agents, while pp8-res50 and pp8-eff are

| Adaptor | Fusion | OPV2V | | | |
| | | Pp8-res50* | | pp8-eff+ | |
| | | AP@0.5 | AP@0.7 | AP@0.5 | AP@0.7 |
|---|---|---|---|---|---|
| MPDA (Xu et al., 2022a) | F-cooper | 69.11 | 53.91 | 66.33 | 44.99 |
| PnPDA (Luo et al., 2024) | | 71.52 | 56.88 | 70.05 | 55.13 |
| Polyinter (Xia et al., 2025) | | 67.07 | 52.84 | 66.89 | 52.20 |
| P-MoE (Ours) | | **71.92** | **57.01** | **70.46** | **56.50** |
| MPDA (Xu et al., 2022a) | CoBEVT | 76.24 | 55.56 | 72.24 | 54.84 |
| PnPDA (Luo et al., 2024) | | 73.49 | 57.14 | 71.68 | 53.18 |
| Polyinter (Xia et al., 2025) | | 72.49 | 56.28 | 70.41 | 53.35 |
| P-MoE (Ours) | | **78.01** | **60.43** | **74.74** | **59.16** |

*Table 2.* Comparison of detection performance of different feature interpreters in different heterogeneous scenarios. We evaluate the performance of different adaptors on the FCooper (Chen et al., 2019) and CoBEVT (Xu et al., 2022c) fusion methods. The symbol "*" denotes models trained with the pp8-res50-vn4 encoder configuration, while the symbol "+" indicates the pp8-eff-vn4 combination. Evaluation is conducted across different heterogeneous settings, including pp8-res50 and pp8-eff.

| Method | OPV2V (3 agents) | | | | DAIR-V2X (2 agents) | | | |
| | pp8-res50-vn4 | | pp8-eff-vn4 | | pp4-res50 | | pp4-eff | |
| | AP@0.5 | AP@0.7 | AP@0.5 | AP@0.7 | AP@0.5 | AP@0.7 | AP@0.5 | AP@0.7 |
|---|---|---|---|---|---|---|---|---|
| F-cooper (Chen et al., 2019) | 71.38 | 53.52 | 70.83 | 54.99 | 63.51 | 45.63 | 60.53 | 44.16 |
| AttFuse (Xu et al., 2022e) | 65.39 | 44.17 | 72.42 | 36.86 | 65.05 | 49.23 | 66.05 | 52.60 |
| V2X-ViT (Xu et al., 2022d) | 73.31 | 50.99 | 76.07 | 55.36 | 65.82 | 48.64 | 67.61 | 49.38 |
| CoBEVT (Xu et al., 2022b) | 75.50 | 56.27 | 75.44 | 56.57 | 66.30 | 45.25 | 66.34 | 44.97 |
| HM-ViT (Xiang et al., 2023) | 72.32 | 51.81 | 71.34 | 47.11 | 67.44 | 50.77 | 67.87 | 51.42 |
| HEAL (Lu et al., 2024) | 73.80 | 54.59 | 72.26 | 53.49 | 67.30 | 53.38 | 67.15 | 53.04 |
| COCMT (Wang et al., 2025) | 75.56 | 55.60 | 73.24 | 55.25 | 67.94 | 53.57 | 67.24 | 52.95 |
| X-MoGe (Ours) | **79.54** | **60.42** | **76.52** | **59.01** | **68.15** | **54.93** | **67.97** | **53.70** |

*Table 3.* Comparison of detection performance of different fusion methods on cross-modal sensors. We adopt the end-to-end training strategy for all fusion methods on both OPV2V and DAIR-V2X datasets. Among all methods, our X-MoGe achieves the best detection performance.

used for evaluation. For DAIR-V2X, which contains only two agents (one vehicle and one infrastructure), we fix the ego agent to pp4 and configure the ego–neb pair as pp4-res50 and pp4-eff, respectively. Evaluation is conducted separately under these two settings.

All models are trained using a batch size of 2 for 40 epochs using the Adam optimizer (initial LR = 0.001, weight decay=1e-4) and a MultiStep scheduler (decay at epochs 15 and 50) on 4 NVIDIA GeForce RTX 3090 GPUs.

### 4.3. Quantitative Evaluation

**Comparison of Detection Performance.**

Table 2 compares different feature interpreters under heterogeneous settings on OPV2V. The proposed P-MoE consistently achieves the best performance across all configurations. Under the F-Cooper (Chen et al., 2019) framework, P-MoE outperforms existing adaptors under different heterogeneous settings, with gains on the AP@0.7 metric (+0.13

and +1.37 AP over PnPDA on pp8-res50 and pp8-eff), indicating improved localization robustness.

When integrated with the stronger CoBEVT (Xu et al., 2022b) fusion method, the performance gains brought by P-MoE become more pronounced. Compared with MPDA (Xu et al., 2022a) and PnPDA (Luo et al., 2024), P-MoE yields substantial improvements across all metrics, particularly under the pp8-eff setting, where it achieves 74.74 AP@0.5 and 59.16 AP@0.7, surpassing the best competing method by a clear margin. This suggests that P-MoE can effectively adapt to heterogeneous feature distributions and better exploit the complementary characteristics of different sensor encoders.

Table 3 compares different fusion methods under cross-modal settings on OPV2V and DAIR-V2X datasets. X-MoGe consistently outperforms existing methods across all configurations. On OPV2V with three agents, X-MoGe achieves 60.42 AP@0.7 under pp8-res50-vn4, improving

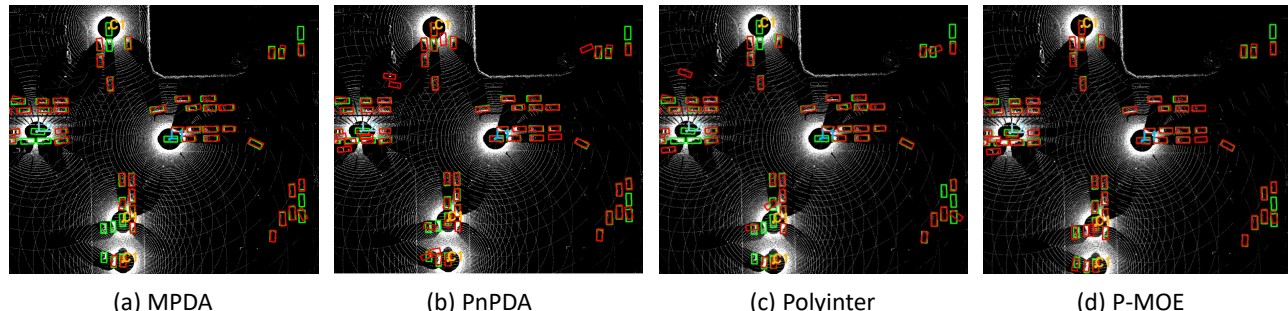

| (a) MPDA | (b) PnPDA | (c) Polyinter | (d) P-MOE |

*Figure 6.* Qualitative comparison results of different feature interpreters on the OPV2V dataset. The green boxes denote the ground truth, while the red boxes indicate the predicted results. We compare the test results under the pp8–res50–vn4 configuration with L1 as the ego vehicle, where L1 corresponds to pp8, L2 corresponds to vn4, and C1 denotes the ResNet-50 encoder.

upon the strongest baseline by +4.15 AP, and reaches 59.01 AP@0.7 under pp8-eff-vn4, exceeding by +2.44 AP. On DAIR-V2X with two agents, X-MoGe attains 54.93 AP@0.7 (pp4-res50) and 53.70 AP@0.7 (pp4-eff), outperforming the best competing methods by +1.36 and +0.66 AP, respectively. These consistent improvements demonstrate the effectiveness of X-MoGe in handling heterogeneous cross-modal feature distributions and achieving more accurate perception.

### 4.4. Qualitative Evaluation

Figure 6 presents the detection results of MPDA (Xu et al., 2022a), PnPDA (Luo et al., 2024), PolyInter (Xia et al., 2025), and P-MoE under the heterogeneous pp8–res50–vn4 setting on the OPV2V dataset. For targets located near the LiDAR-based agents L1 and L2, P-MoE produces detection results that are largely comparable to those of previous methods. In contrast, for targets in the vicinity of the image-based agent C1, P-MoE yields bounding boxes that align more accurately and comprehensively with the ground truth. This observation indicates that P-MoE is more effective at mitigating cross-modal semantic discrepancies and improving feature interpretation for camera-based agents in heterogeneous collaborative perception scenarios.

### 4.5. Ablation Studies

Table 4 reports an ablation study on DAIR-V2X (pp4-res50), where ID=5 corresponds to the full model.

Compared with the full model (ID=5), removing the MoE adaptor (ID=1) leads to a significant drop of 2.87 AP@0.5 and 3.82 AP@0.7, indicating that pixel-wise expert selection is critical for handling heterogeneous feature distributions. When the UA-head is removed (ID=4), performance decreases by 0.77 AP@0.5 and 0.57 AP@0.7, suggesting that uncertainty-aware fusion helps stabilize predictions under cross-agent noise. Excluding EAE (ID=3) mainly affects high-IoU performance, with AP@0.7 dropping by 1.32, im-

| | Adaptor | Fusion | | | DAIR-V2X (pp4-res50) | | |
|---|---|---|---|---|---|---|---|
| ID | P-MoE | SGE | EAE | UA-head | AP@0.3 | AP@0.5 | AP@0.7 |
| 1 | | ✓ | ✓ | ✓ | 70.63 | 65.28 | 51.11 |
| 2 | ✓ | | ✓ | ✓ | 72.03 | 66.78 | 52.11 |
| 3 | ✓ | ✓ | | ✓ | 71.79 | 66.57 | 53.61 |
| 4 | ✓ | ✓ | ✓ | | 72.74 | 67.38 | 54.36 |
| 5 | ✓ | ✓ | ✓ | ✓ | **74.22** | **68.15** | **54.93** |

*Table 4.* Evaluation of the effectiveness of different components. We compared the accuracy of pp4-res50 in the DAIR-V2X dataset after sequentially removing the designed modules.

plying that edge-aware enhancement contributes to precise localization. Similarly, removing SGE (ID=2) results in a performance degradation of 1.37 AP@0.5 and 2.82 AP@0.7, highlighting the importance of geometry-aware modeling for accurate object alignment.

The performance drops observed when removing each component confirm that P-MoE, SGE, EAE, and UA-head are all indispensable and complementary, and jointly enable robust detection under heterogeneous collaborative perception settings.

## 5. Conclusion

In this paper, we propose X-MoGe, a cross-modal adaptation framework for heterogeneous multi-agent collaborative perception. By combining a pixel-level Mixture-of-Experts module for modality-aware semantic adaptation with a geometry-guided fusion mechanism for BEV alignment, X-MoGe effectively addresses semantic and geometric inconsistencies across heterogeneous agents. Experimental results on OPV2V and DAIR-V2X demonstrate that the proposed method achieves superior performance and robustness compared to existing approaches, highlighting its effectiveness and scalability in real-world heterogeneous collaborative perception scenarios.

## Impact Statement

This paper presents work whose goal is to advance the field of Machine Learning. There are many potential societal consequences of our work, none which we feel must be specifically highlighted here.

## Acknowledgement

This work was supported by the National Natural Science Foundation of China (No.42571514).

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
