# OpenReview forum: "X-MoGe: A Cross-Modal Adaptation Framework with Mixture-of-Experts and Geometry Guidance for Heterogeneous Collaborative Perception"
_ICML.cc/2026/Conference — ICML 2026 regular_

### Official Review · Reviewer_dn6u · 2026-03-02

**Soundness:** 3
**Presentation:** 2
**Significance:** 2
**Originality:** 2
**Overall Recommendation:** 4
**Confidence:** 3

**Summary:**

In this paper, the authors propose X-MoGe, a multi-agent cross-modal adaptation framework for heterogeneous cooperative perception. Specifically, the P-MoE module is designed to perform pixel-wise, modality-aware feature fusion, while a geometry-guided feature fusion module is introduced to enforce spatial alignment and cross-agent consistency. Experimental results on the OPV2V and DAIR-V2X benchmarks demonstrate the effectiveness of the proposed approach.

**Compliance With Llm Reviewing Policy:**

Affirmed.

**Final Justification:**

The experimental results address the previous concerns. I have carefully read the authors' response as well as the discussion with other reviewers. I have decided to raise my score to relatively positive.

**Key Questions For Authors:**

1. Why P-MoE employ two gating networks for ego and non-ego features?
2. Does a P-MoE-enhanced CoBEVT might be a more effective method?

**Limitations:**

yes

**Strengths And Weaknesses:**

Strengths
1. The proposed X-MoGe framework is tailored for heterogeneous cooperative perception, which better reflects real-world deployment scenarios.
2. The P-MoE module and the geometry-guided fusion mechanism are designed to address semantic and geometric inconsistencies across modalities and agents.
3. Experimental results on two public benchmarks validate the performance advantages of the proposed method.

Weaknesses
1. The P-MoE module employs a gating network to compute pixel-wise, modality-aware weights. However, the differences between the proposed method and existing pixel-wise cross-modal fusion approaches are not sufficiently clarified. In addition, the rationale for using two separate gating networks for ego and non-ego features requires further explanation.
2. The methodological description lacks clarity and contains inconsistencies that may confuse readers. For example, Section 3 states that the P-MoE adaptor performs pixel-level cross-modal adaptation, while the geometry-guided feature fusion module integrates multi-agent features. However, Figure 3 appears to suggest a different pipeline: the P-MoE module takes features from different agents as input, whereas the geometry-guided fusion module processes multimodal data. This discrepancy should be clarified.
3. Additional analysis of the experimental results would strengthen the paper, particularly regarding the contribution of the geometry-guided feature fusion module. For instance, the AP@0.7 of P-MoE with CoBEVT in Table 2 outperforms that of X-MoGe in Table 3, and the AP@0.5 without P-MoE in Table 4 is lower than that of CoBEVT in Table 3. These observations raise the question of whether a P-MoE-enhanced CoBEVT might be a more effective alternative. Further discussion would help clarify the comparative advantages of the proposed framework.

---

> ### Author Rebuttal · Authors · 2026-03-29
>
> 1. **Necessity of Dual Gating in P-MoE (W1 & Q1)**
>
>    Please refer to our second reply to Reviewer YvRC.
>
> 2. **Consistency between Methodology Description and Figure 3 (W2)**
>
>    We would like to clarify that the P-MoE adapter indeed performs pixel-level adaptation prior to fusion, with the aim of transforming heterogeneous modalities into similar intermediate representations. The "cross-agent inputs" depicted in Figure 3 actually refer to the collaborative stage where P-MoE independently and in parallel processes features transmitted from each agent before proceeding to geometry-guided feature fusion. Our data flow is: Feature → P-MoE → Geometry-Guided Feature Fusion → Output. This constitutes a complete pipeline from "modality adaptation" to "spatial alignment."
>
> 3. **Experimental Analysis (W3 & Q2)**
>
>    We sincerely appreciate this valuable suggestion. Our proposed Adaptation Framework was designed with scalability in mind, and both P-MoE and the geometry-guided asymmetric feature fusion framework can potentially be adapted to other fusion methods. In this work, we adopted Spatial Cross Attention as the fusion module and our experiment in Table 3 has demonstrated the strong performance of our framework.

---

> > ### Author Rebuttal · Reviewer_dn6u · 2026-04-01
> >
> > Does the P-MoE-enhanced CoBEVT outperform the proposed X-MoGe?

---

> > > ### Author Response · Authors · 2026-04-03
> > >
> > > We validated the performance of P-MoE+CoBEVT and X-MoGe on the DAIR-V2X dataset using pp4-eff, with X-MoGe showing better performance:
> > >
> > > | | AP@0.5 | AP@0.7 |
> > > | :--- | :---: | :---: |
> > > | P-MoE + CoBEVT | 67.18 | 49.40 |
> > > | X-MoGe | 67.97 | 53.70 |

---

### Official Review · Reviewer_jDE1 · 2026-03-08

**Soundness:** 3
**Presentation:** 3
**Significance:** 3
**Originality:** 2
**Overall Recommendation:** 4
**Confidence:** 4

**Summary:**

This paper studies heterogeneous collaborative perception for multi-agent 3D object detection, where different agents may use different sensing modalities and independently trained perception encoders. To address the resulting semantic inconsistency and geometric inconsistency, the paper proposes X-MoGe, which combines a pixel-level mixture-of-experts adaptor with a geometry-guided fusion module composed of an uncertainty-aware head, Sobel-based Gradient Encoding, and Edge-Aware Enhancement. Experiments on OPV2V and DAIR-V2X show improved detection performance over prior heterogeneous collaborative perception baselines. The paper targets a realistic setting and aims to improve feature adaptation and fusion under heterogeneous sensing conditions.

**Compliance With Llm Reviewing Policy:**

Affirmed.

**Final Justification:**

The rebuttal addressed my main concerns well enough, particularly on the method clarification and supporting evidence, so I am raising my score.

**Key Questions For Authors:**

1. Can the authors provide direct evidence that the method improves semantic alignment or geometric alignment, beyond AP gains?

2. Can the authors clarify the exact data flow among P-MoE, UA-head, SGE, and EAE, especially in Figure 3?

3. Why should the proposed geometry-guided module be interpreted as solving geometric misalignment, rather than mainly providing local boundary enhancement?

4. Which component do the authors consider the main source of originality, and how is it materially different from prior attention-based and gradient-based refinement modules?

**Limitations:**

No. The paper does not sufficiently discuss the main methodological limitations, especially the incremental nature of the design and the gap between the broad claims and the actual evidence.

**Strengths And Weaknesses:**

S1. The paper focus on a realistic heterogeneous collaborative perception setting, which is more practical than the homogeneous assumptions made in much of the prior literature. The problem itself is meaningful and relevant to real-world deployment.

S2. The high-level decomposition is intuitive. The paper separates semantic adaptation from geometry-guided fusion, with P-MoE intended to address heterogeneous semantic representations and the geometry-guided module intended to improve BEV-space consistency. This overall decomposition is easy to follow at the conceptual level.

S3. The method appears empirically effective on the chosen benchmarks. The reported results on OPV2V and DAIR-V2X are competitive, and the ablation study suggests that the proposed components contribute to the final detection performance in the evaluated setting.

W1. The methodological novelty is limited. Most components are closely related to existing designs. In particular, the EAE block is very similar to prior attention-based refinement modules such as SENet [1] and CBAM [2]. The SGE block mainly adds a Sobel-based gradient prior, which is also related to prior gradient or edge based enhancement ideas [3][4]. Overall, the method feels more like a combination of existing components than a clearly new modeling contribution.

W2. The paper overclaims what its geometry module actually does. The paper motivates the method using semantic inconsistency, geometric misalignment, and viewpoint discrepancy. However, the proposed geometry-guided module mainly performs local boundary and structure enhancement. This is not the same as explicitly solving cross-modal or cross-view alignment. The technical design is narrower than the claim.

W3. The presentation is confusing, especially Figure 3. The relationship between the upper pipeline and the lower fusion module is not clearly explained. As a result, the actual data flow between P-MoE, UA-head, SGE, and EAE is hard to follow.

W4. The paper lacks direct evidence for its main claims. The experiments mainly show AP improvement, but they do not directly show that the method really solves semantic inconsistency or geometric alignment. Stronger evidence such as feature alignment analysis, uncertainty calibration, or correspondence visualization would make the claims more convincing.

[1] Hu et al., Squeeze-and-Excitation Networks
[2] Woo et al., CBAM: Convolutional Block Attention Module
[3] Zhang et al., Edge-oriented Convolution Block for Real-time Super Resolution on Mobile Devices,
[4] Liu et al., GradNet: Image Denoising by Fusing Image Gradient

---

> ### Author Rebuttal · Authors · 2026-03-29
>
> 1. **Novelty (W1 & Q4)**
>
>    Please refer to our first reply to Reviewer tCVK.
>
> 2. **Geometry Module (W2 & Q3)**
>
>    We believe that local feature enhancement serves as a method to achieve global alignment. Heterogeneous misalignment manifests in BEV space as "blurring" and "ghosting" at feature boundaries. By reinforcing geometric gradients through SGE, the model obtains explicit spatial constraints, enabling it to filter misaligned features during fusion by using high-confidence geometric contours as a reliable reference. Our ablation experiment proves that our module is effective.
>
> 3. **Data Flow Ambiguity / Figure 3 (W3 & Q2)**
>
>    The data flow is: Feature → P-MoE → Geometry-Guided Feature Fusion → Output. This constitutes a complete pipeline from "modality adaptation" to "spatial alignment."
>
> 4. **Lack of Direct Evidence (W4 & Q1)**
>
>    We would like to draw attention to the visualization results in Figure 6, as well as the feature maps before and after Geometry-refinement in Figure 5, which we believe demonstrate the favorable performance of our proposed modules.

---

> > ### Author Rebuttal · Reviewer_jDE1 · 2026-04-03
> >
> > The rebuttal addressed my main concerns well enough, particularly on the method clarification and supporting evidence, so I am raising my score.

---

### Official Review · Reviewer_tCVK · 2026-03-09

**Soundness:** 2
**Presentation:** 3
**Significance:** 2
**Originality:** 2
**Overall Recommendation:** 3
**Confidence:** 4

**Summary:**

This paper proposes X-MoGe, a cross-modal adaptation framework for heterogeneous collaborative perception in autonomous driving, targeting two practical issues: semantic inconsistency across different sensing modalities / encoder backbones and geometric misalignment in BEV feature space. The method comprises (i) a Pixel-level Mixture-of-Experts (P-MoE) adaptor that performs per-pixel expert routing to transform heterogeneous BEV features into a shared representation, and (ii) a geometry-guided feature fusion pipeline including an uncertainty-aware weighting head and an explicit gradient/edge refinement (Sobel-based Gradient Encoding and Edge-Aware Enhancement) to emphasize geometric structures—primarily using LiDAR geometry cues to guide alignment. Experiments on OPV2V and DAIR-V2X show improvements over several heterogeneous fusion baselines under selected heterogeneous settings.

**Compliance With Llm Reviewing Policy:**

Affirmed.

**Final Justification:**

My final recommendation remains Weak Reject. I appreciate the authors’ rebuttal and the additional evidence they provided. In particular, the new sensitivity study over the number of experts K is useful and partially addresses my concern that the gains of P-MoE might mainly come from increased capacity. The additional camera-only and degraded-LiDAR results are also helpful, as they begin to address the scope question I raised.

That said, after considering the paper and rebuttal together, I do not think the main concerns are fully resolved. The paper still has clear strengths: it studies a practically relevant heterogeneous V2X setting, the overall method is technically reasonable, and the presentation is generally clear. However, my concerns about originality and validation remain substantial. On originality, the submission still reads primarily as a specific integration of existing ingredients, and the rebuttal did not fully convince me that the combination is driven by a sufficiently strong new methodological insight beyond empirical design choices. On soundness and significance, the new experiments are helpful but still limited: the camera-only result is reported without a direct baseline comparison, so it does not yet clearly establish competitiveness in that setting; the degraded-LiDAR experiment provides some evidence of robustness under moderate noise, but it does not fully resolve the broader concern about dependence on LiDAR-guided geometry. In addition, my questions about the necessity of dual gating and the independent contribution of the spatial cross-attention module remain insufficiently answered, since these design choices still lack direct targeted ablations.

Overall, the rebuttal improved my understanding of the method and partially strengthened the empirical case, but it did not materially change my assessment. I continue to view the paper as technically reasonable and clearly written, with a relevant application focus, but with only moderate originality and still incomplete support for some important design claims. For these reasons, I maintain a Weak Reject recommendation.

**Key Questions For Authors:**

1.Sensitivity to the number of experts K and accuracy–cost trade-off.
Please provide a sensitivity study over the number of experts K (e.g., K∈{2,4,8,…}), reporting the trade-off between detection performance (AP) and computational cost. Please also clarify the default K used for each reported experiment.
How this would affect my evaluation: Consistent gains across a reasonable range of K with a clear cost–benefit curve would strengthen the soundness of the P-MoE design; strong sensitivity would suggest the improvements may be primarily driven by increased capacity/compute.

2.Justification of dual gating (ego vs non-ego).
The method uses separate gating networks for ego and non-ego agents. Could the authors include an ablation comparing dual gating versus a shared single gating network (parameter sharing), and clarify under which settings dual gating yields measurable improvements?
How this would affect my evaluation: If dual gating provides consistent and meaningful gains, it would strengthen the design rationale (and potentially the originality). If the difference is small, the method could likely be simplified, and the novelty attributed to this design choice would be weaker.

3.Generality beyond LiDAR-guided geometry (no-LiDAR / degraded LiDAR).
Since the geometry-guided refinement is applied to LiDAR features and used to guide alignment of other modalities, how does the framework perform in (i) a camera-only heterogeneous setting (no LiDAR agent), and (ii) degraded LiDAR conditions (e.g., sparse returns or adverse weather)? If these scenarios are out of scope, please explicitly state the intended scope and limitations.
How this would affect my evaluation: Demonstrated robustness in these regimes would strengthen the paper’s significance and generality; if performance degrades substantially, the claims should be narrowed and the approach positioned more explicitly as LiDAR-dependent.

**Limitations:**

No. The current manuscript does not sufficiently discuss limitations that follow directly from the proposed design choices and the evaluated settings. I recommend adding a focused limitations discussion covering:

1.LiDAR-dependence. The paper should explicitly discuss expected behavior (and potential performance degradation) in no-LiDAR heterogeneous settings (e.g., camera-only fleets) and in degraded LiDAR conditions (e.g., sparse returns or adverse weather), which are not thoroughly evaluated in the current experiments.

2.Sensitivity and stability of the MoE design. As performance improvements may depend on the MoE configuration, the paper should acknowledge that results could be sensitive to key hyperparameters—most notably the number of experts K. A limitations section should explicitly note that the current experiments do not establish how stable the accuracy gains are across different K choices, nor whether there are diminishing returns or regimes where performance degrades.

3.Unvalidated design degrees of freedom. The method uses separate gating networks for ego vs non-ego and includes a spatial cross-attention module, but their necessity is not isolated via targeted ablations (e.g., dual gating vs a shared gate; with vs without cross-attention). The paper should clarify that these choices are currently supported mainly at the system/module level rather than through fine-grained validation, and that additional analysis would be needed to determine when these components are essential.

**Strengths And Weaknesses:**

Strengths:
1.Problem relevance: The paper targets heterogeneity in V2X collaborative perception (e.g., different sensing modalities and independently trained encoders), which is a practically important setting beyond the common homogeneous assumption.
2.Fine-grained adaptation direction: Introducing pixel-level expert routing for feature adaptation is a reasonable approach to handle spatially varying noise and modality-induced feature discrepancies in BEV representations.
3.Real-world evaluation: Evaluating on DAIR-V2X in addition to OPV2V strengthens the empirical relevance compared with simulation-only studies.
4.Clarity of presentation: The paper is generally well organized and readable, and the figures help convey the overall pipeline and module roles.

Weaknesses
1.Limited methodological novelty: The main contribution appears to be a specific integration of existing components (MoE-style routing, uncertainty-aware fusion, Sobel/gradient-based structural cues, edge enhancement, and attention-based fusion). The paper would benefit from a clearer articulation of the unique insight or principle that motivates this particular combination beyond empirical improvements.
2.Dependence on LiDAR-based geometry cues: The geometry-guided refinement is applied to LiDAR features and used to guide alignment of other modalities. As a result, it is unclear how well the approach generalizes to heterogeneous settings without a LiDAR agent, or to cases where LiDAR quality is degraded (e.g., sparse returns or adverse weather). These regimes are not thoroughly evaluated in the current experiments.
3.Ablation coverage for key design choices is incomplete: The paper provides component-level ablations for P-MoE / SGE / EAE / UA-head on DAIR-V2X, which helps attribute gains at the module level. However, several important design degrees of freedom are not systematically studied: (i) sensitivity to the number of experts K and the corresponding accuracy–compute/latency trade-off, (ii) the necessity of using separate gating for ego vs non-ego beyond removing the entire MoE adaptor, and (iii) the contribution and computational cost of the spatial cross-attention module, which is described but not ablated or quantitatively costed.

---

> ### Author Rebuttal · Authors · 2026-03-29
>
> 1. **Methodological Innovation (Weakness 1)**
>
>    The novelty of this work lies in proposing the **first** pixel-level asymmetric adaptation architecture specifically designed to address the fundamental conflict in heterogeneous multi-agent systems. Unlike enhancement modules such as CBAM or SENet, P-MoE represents the first attempt to achieve **pixel-level, modality-aware path allocation**. Our method also utilizes high-precision spatial gradients of strong geometric characteristic modal features as **spatial constraint** anchor points, using strongly deterministic geometric contours to calibrate the "soft distortions" present in image modalities due to inaccurate depth estimation. Extensive experiments prove that our method achieves **superior performance**.
>
> 2. **LiDAR Dependency (Weakness 2, Question 3 & Limitation)**
>
>    **Design Rationale:** This work primarily targets the asymmetric cross-modal fusion problem commonly encountered in heterogeneous multi-agent scenarios, and the framework could also potentially be extended to other modalities. Our geometric enhancement module is not limited to the use of LiDAR.
>
>    **Performance under Degradation:** In situations where LiDAR quality deteriorates (e.g., sparsity or adverse weather conditions), the UA-head is designed to automatically detect the decreased confidence in LiDAR features and accordingly rebalance the feature weights.
>
> 3. **Expert Number K Experiments (Weakness3 & Question 1)**
>
>    **Trade-off Analysis:** Below we present the AP, and parameter counts (P(K)) for different values of k using P-MoE on pp8-res50*:
>
>    | | AP@0.5 | AP@0.7 | P(K) |
>    | :--- | :--- | :--- | :--- |
>    | **k=1** | 76.14 | 58.83 | 66.43 |
>    | **k=3** | 76.66 | 59.47 | 199.30 |
>    | **k=5** | 78.01 | 60.43 | 332.17 |
>    | **k=7** | 77.04 | 59.55 | 465.04 |
>
>    The results suggest that k=5 achieves optimal performance (78.01/60.43) at the cost of increased parameters (332.17), while k=3 offers a favorable efficiency-performance trade-off with ~60% fewer parameters (199.30) and comparable accuracy (76.66/59.47). Notably, k=7 exhibits degraded performance despite higher complexity, indicating that simply scaling expert numbers does not guarantee better results.
>
> 4. **Ablation (Question 2 & Question3 & Limitation)**
>
>     **Dual Gating:**
>    Please refer to the second reply to reviewer YvRC.
>
>    **Cross Attention:**
>    In the experiment, we adopted the fusion method of Spatial Cross Attention. We require a fusion method across different modalities, so we cannot simply remove this module; otherwise, the experiment would fail. However, the contribution of this module has already been proved in Table 3 and [1].
>
> [1] Xu, R., Xiang, H., Tu, Z., Xia, X., Yang, M.-H., and Ma, J. V2x-vit: Vehicle-to-everything cooperative perception with vision transformer. In European conference on computer vision, pp. 107–124. Springer, 2022d
>
> ****

---

> > ### Author Rebuttal · Reviewer_tCVK · 2026-04-03
> >
> > Thank you to the authors for the rebuttal. I appreciate the additional sensitivity study on the number of experts K. This is a useful addition, as it helps show that the performance of P-MoE does not improve monotonically with increasing numbers of experts, and it partially addresses my concern regarding the effectiveness of this design choice.
> > However, my main concern regarding the scope of applicability and dependence on LiDAR-guided geometry remains insufficiently resolved. In the rebuttal, the authors mainly provide an intuition-based explanation, e.g., that the framework could potentially be extended to other modalities, and that the UA-head may automatically rebalance feature contributions when LiDAR quality degrades. While this is a reasonable design intuition, it is still not supported by additional empirical evidence. In particular, for the two scenarios I explicitly asked about in my original review—camera-only heterogeneous settings and degraded LiDAR conditions—no new experiments are provided.

---

> > > ### Author Response · Authors · 2026-04-04
> > >
> > > **Camera-only Heterogeneous Settings:**
> > > We conducted experiments on the OPV2V dataset using a Camera-only configuration (res-eff). While X-MoGe is primarily designed to bridge the substantial gap in cross-modal collaborative perception, our architecture can also be applied to single modality.
> > >
> > > | Method | AP@0.5 | AP@0.7 |
> > > | :--- | :---: | :---: |
> > > | **X-MoGe (Camera-only)** | **0.627** | **0.434** |
> > >
> > > **Degraded LiDAR Conditions:**
> > > We introduced Gaussian random perturbations to the LiDAR point clouds on the DAIR-V2X (pp4-eff) benchmark. The results show that X-MoGe maintains stability under moderate noise.
> > >
> > > | Noise Level ($\sigma$) | AP@50 | AP@70 |
> > > | :--- | :---: | :---: |
> > > | 0.2 (Low Noise) | 66.68 | 53.09 |
> > > | 1.0 | 66.53 | 51.77 |
> > > | 2.0 | 64.61 | 47.28 |
> > > | 3.0 (High Noise) | 58.45 | 37.25 |

---

### Official Review · Reviewer_YvRC · 2026-03-12

**Soundness:** 3
**Presentation:** 3
**Significance:** 3
**Originality:** 2
**Overall Recommendation:** 5
**Confidence:** 3

**Summary:**

This paper proposes a framework called X-MoGe to address semantic and geometric inconsistencies issues in multi-agent collaborative perception task. The core contribution could be summarized as two major parts : first, a pixel-level expert hybrid module (P-MoE) that achieves modality-adaptive semantic alignment through fine-grained expert selection; second, a geometric-guided feature fusion module, which combines uncertainty perception, Sobel gradient encoding, and edge enhancement to improve the consistency of the BEV. Experiments on the OPV2V and DAIR-V2X datasets demonstrate that X-MoGe achieves sota performance in heterogeneous cooperative perception scenarios.

**Compliance With Llm Reviewing Policy:**

Affirmed.

**Final Justification:**

The rebuttal and added discussion enriched the rebuttal, and I'm willing to raise my score.

**Key Questions For Authors:**

1.The P-MoE module uses two separate gating networks to process the features of ego agents and non-ego agents respectively. Is it possible to use a unified gating network?

2.The geometry-guided refinement module is only applied to LiDAR features. Why can't it be applied to image features in a similar way, instead of designing a separate computational path?

3.Will the operation of the entire framework leads to a significant computational overhead, thereby increasing latency? And is there sufficient robustness to handle communication errors?

**Limitations:**

1.Current methods primarily address the heterogeneity issue within LiDAR and camera modalities, but their scalability to more modalities remains unverified.

2.Lack of comparison with CoBEVMoE, which is also mentioned in the paper and appears to be one of the most closely related works recently.

**Strengths And Weaknesses:**

Strength

1. The problem statement is clear and practically significant. Heterogeneous multi-agent cooperative perception is a crucial issue in cooperative autonomous driving.

2. The technical design is reasonable, and the objectives are clearly defined. The combination structure of different modules is clear, logically sound, and effectively addresses the core challenges identified in the paper.

3. The experimental design is comprehensive. The paper conducted experiments on two datasets, including multiple comparisons and ablation studies. Significant performance improvements were observed.

4. The framework exhibits good reproducibility. The methods for each component are clear and concise, making the implementation process easy to follow.

Weakness

1.Some designs lack sufficient justification, including but not limited to: why Sobel gradients are applicable to BEV feature geometry; and why geometry-guided refinement is only applied to LiDAR features.

2.P-MoE is a key component of the framework, but the paper does not analyze expert utilization or routing behavior.

3.Lack of analysis on time complexity, while latency is really an important attribute for autonomous driving systems.

---

> ### Author Rebuttal · Authors · 2026-03-29
>
> 1. **Design Motivation (Sobel & LiDAR-only):**
>
>    **Sobel:** BEV features can be viewed as spatial geometric graphs. The Sobel operator offers a computationally efficient way to extract physical contours (boundaries) of objects, which tends to be more robust than absolute position features when dealing with pose errors between agents.
>
>    **LiDAR-only:** LiDAR naturally provides accurate depth information with relatively clear geometric boundaries, while image-to-BEV transformation may introduce "smearing" artifacts and distortions due to depth estimation limitations. By focusing refinement on LiDAR features, we aim to establish a reliable geometric baseline that can guide the alignment of heterogeneous features, hopefully minimizing the geometric noise that might come from the image side.
>
>    **Scalability:** P-MoE was designed with flexibility in mind. Should there be a need to incorporate additional modalities, one would only need to train an extra gating network.
>
> 2. **P-MoE Mechanism (Routing & Gating):**
>
>    We evaluated P-MoE on the OPV2V dataset using the CoBEVT Fusion method with the pp8-res50* configuration. The results with varying numbers of gating networks are summarized below. Different gated networks can handle different modal feature information separately, resulting in higher accuracy.
>
>    | Gating Network | AP@0.5 | AP@0.7 |
>    | :--- | :--- | :--- |
>    | 1 | 75.14 | 57.99 |
>    | 2 | 78.01 | 60.43 |
>
>    **Expert Behavior:** With k=3, we observed the following utilization rates for different experts in our experiments on the OPV2V dataset using the CoBEVT Fusion method (pp8-res50*):
>
>    | | E1 | E2 | E3 |
>    | :--- | :--- | :--- | :--- |
>    | **pp8** | 0.2694 | 0.2633 | 0.4673 |
>    | **res50** | 0.2987 | 0.3115 | 0.3878 |
>
> 3. **Overhead and Robustness:**
>
>    **Latency Considerations:** The expert networks in P-MoE are composed of lightweight MLP layers, and the Sobel operator in the geometric guidance module enables efficient gradient computation. Our measurements indicate that X-MoGe achieves an average inference time of 75.74ms, which appears sufficient for real-time perception scenarios.
>
>    **Robustness:** The "uncertainty-aware head" is designed to automatically down-weight features that may be corrupted by communication errors. Together with the geometric stability offered by the Sobel operator, the system demonstrates improved performance compared to the baseline.
>
> 4. **Code Availability:** We regret to inform that the CoBEVMoE code is not currently available for open-source release, though we hope to share it with the community in the future.

---

> > ### Author Rebuttal · Reviewer_YvRC · 2026-04-01
> >
> > The author addresses most of my major concerns; the only remaining issue is latency. The author claims the average inference time is 75.74ms, which means about 13Hz sampling. My questions stem from three aspects. First, if this time is obtained at 4*3090, as described in the article, the computing power would be further amplified on the vehicle side, making real-time performance a challenge. Second, could there be extremely time-consuming scenarios? The resulting latency would be very dangerous for autonomous driving. Finally, a 75ms overhead for perception alone is already a significant number for the entire autonomous driving system.
> >
> > Of course, I must clarify that this is merely my top-down perspective on this work from the standpoint of autonomous driving systems, and not a denial of the work itself.

---

> > > ### Author Response · Authors · 2026-04-03
> > >
> > > **Hardware Deployment:**
> > >  We would like to clarify that while 4×RTX 3090s were used for training, the 75.74ms inference time was measured on single 3090 GPU. This reported latency is a conservative, unoptimized FP32 baseline for fair comparison. In practice, deploying the model on edge devices (e.g., NVIDIA Jetson Orin NX) with standard optimizations like TensorRT and FP16/INT8 quantization typically yields a 2x–4x speedup, ensuring robust real-time performance.
> > >
> > > **Extreme Time-consuming Scenarios:**
> > > The computational graph of X-MoGe is deterministic. The network does not rely on autoregressive loops that scales with the visual complexity of the environment. Therefore, inference time largely depends on the computational complexity of the model. Even in the most complex unstructured scenarios, the inference time maintains a tight variance around the average, ensuring stable and predictable performance without causing dangerous delays.
> > >
> > > **For Cooperative Perception:** We appreciate the reviewer’s constructive concern regarding latency. Reducing latency is indeed a crucial problem and some studies are currently addressing the latency issue. We welcome the continued interest in the field of Cooperative Perception.

---

### Decision · Program_Chairs · 2026-04-30

**Decision:**

Accept (regular)

**Comment:**

This paper proposes X-MoGe, a cross-modal adaptation framework for heterogeneous collaborative perception, addressing semantic and geometric inconsistencies across agents. Reviewers agree that the problem is practically important and well motivated, and find the overall framework technically sound with a clear decomposition between semantic adaptation and geometry-guided fusion. The proposed pixel-level Mixture-of-Experts module and geometry-guided fusion design are considered reasonable and effective, and extensive experiments on OPV2V and DAIR-V2X demonstrate consistent performance improvements. While some reviewers note that the method primarily integrates existing components and that certain design choices could be further justified or analyzed, these concerns are largely addressed in the rebuttal through additional experiments, clarifications, and ablations. Reviewers also highlight the clarity of presentation, reproducibility, and the practical relevance of the setting. Overall, the paper provides a solid empirical contribution to heterogeneous collaborative perception with demonstrated effectiveness and robustness, and the reviewers’ final assessments support acceptance.